# Storm Water Management of Low Impact Development in Urban Areas Based on SWMM

**Yiran Bai, Na Zhao \*, Ruoyu Zhang and Xiaofan Zeng**

School of Hydropower and Information Engineering, Huazhong University of Science and Technology, Wuhan 430074, China; m201773764@hust.edu.cn (Y.B.); zhangry@hust.edu.cn (R.Z.); zengxiaofan@hust.edu.cn (X.Z.)
**\*** Correspondence: na.zhao.2011@hust.edu.cn; Tel.: +86-135-5414-7104

**Abstract:** LID (low impact development) is the storm management technique designed for controlling runoff in urban areas, which can be used to solve urban flooding disasters. Taking Sucheng District of Suqian City, Jiangsu Province, China as an example, this project used SWMM (storm water management model) to study the effect of four different types of LID scenarios (① no LID technique, ② LID technique based on infiltration, ③ LID technique based on water storage, ④ LID technique based on the combination of infiltration and water storage) on urban flooding under different rainfall patterns. For the whole study area, the results show that infiltration facilities have the greater reduction rate of surface runoff compared with storage facilities. The combined model (infiltration + storage) works best in the reduction of peak flow and flood volume, with the maximum reduction rate of peak flow (32.5%), and the maximum reduction rate of flood volume (31.8%). For local nodes, infiltration facilities and water storage facilities have different effects. Infiltration facilities significantly reduce runoff of node 47, the reduction rate of ponding time ranges from 73.1% to 54.5%, while water storage facilities have no effects on it. Storage facilities significantly reduce runoff of node 52, the reduction rate of ponding time is 100%, while infiltration facilities have no effects on it. Under all the LID designs, runoff reduction gradually increases with the increasing rainfall amount, and peak reduction becomes stable when rainfall amount reaches about 81.8 mm. In general, the combined model (infiltration + storage) performs better than any other scenarios in runoff reduction. The research shows that LID facilities can greatly mitigate flood, thus the urban flooding disasters caused by extreme rainstorms can be prevented.

**Keywords:** LID; storm management; SWMM; infiltration facilities; water storage facilities

---

## 1. Introduction

With global warming and frequent heavy rains, urban flooding is becoming the serious urban disease in many countries all over the world [1]. In recent years, many cities in China have experienced different levels of flooding disasters, the long time and deep depth of flooding along with large flooding areas are becoming three critical problems demanding solutions [1]. Among 672 cities in mainland China, there are 643 cities with flood control tasks, but only 177 cities have reached national flood control standards, accounting for only 27.6%. The urban flooding disaster has become a common concern in China, and the studies on corresponding simulation technologies have attracted widespread attention [2–4].

The urban flooding problem arises from three aspects as follows. Firstly, in urban areas, rainwater is mainly removed through the rainwater pipe network system. However, in quite a number of cities, low level drainage standards, poor drainage capacity, aged drainage facilities and intricate drainage pipe networks with unreasonable settings, result in severe 'intestinal obstruction'. It is difficult to drain

logged water out of the system in time, thus aggravating urban flooding. Taking Beijing as an example, the drainage pipe network infrastructure in some areas dates back to Ming Dynasty, it is very difficult to fulfill the requirement of draining excessive flood in modern time. Secondly, the area occupied by impermeable pavements and buildings are increasing with the expansion of urban built-up area, blocking the water which is intended to infiltrate into the underground, thus causing flooding on the Earth's surface everywhere. Thirdly, the depressions, lakes, and reservoirs, which are functioning as natural water storage facilities, are gradually filled or destroyed artificially for other purposes. It reduces the performance of these storage facilities in storing and delaying flooding, and mitigates the flood diversion function [5].

Being faced with the above problems, it is challenging to build a modern urban drainage system. Researchers have been dedicated to this filed for over two decades [6]. "Sponge City" has been proposed for realizing the illusion of "natural storage, natural infiltration and natural purification", it devotes to building a benign hydrological cycle system and maximizing the storage, infiltration and purification of rainwater in urban areas [6]. Therefore, the total amount of runoff discharge can be effectively controlled to a low level, and the flood hydrograph with steep rise and fall can be flatted, making to minimizing a series of adverse effects brought about by urbanization. "Sponge City" can not only alleviate the urban flooding disaster in the city, but also be pollution-free and economic, thus drawing more and more attention in recent years.

The concept of "Sponge City" comes from the research of the adsorption effect of the city on the surrounding rural population proposed by Australian scholars, which is referred to the next generation of urban stormwater management technique [7]. The "Sponge City" is also explained as "the low impact development rainwater system construction". It absorbs, infiltrates, purifies, and stores water in raining period of time, and releases the stored water when necessary. In terms of the design of the "Sponge City", researchers in the United States proposed the "Low Impact Development (LID)" [8], which is the stormwater management and non-point source pollution treatment technique developed in late 1990s. It aims to control the runoff and pollution by means of decentralized, small-scale source control, making the development area being close to the natural hydrological cycle pattern as far as possible. LID is a kind of ecological technique which easily realizes urban rainwater collection and utilization. LID facilities mainly include rain gardens, green roofs, permeable pavements, vegetative swales, and bio-retention cells.

LID facilities have been deployed in many application scenarios across developed districts and countries. In the United States, the application of LID facilities has given rise to many ecological construction theories and methods such as green roads, green communities, etc. [9]. In Australia, LID technique is known as water-sensitive urban design [10]. In the UK, the application of LID technique in urban drainage system facilitates the implementation of a sustainable urban drainage system [11]. In Canada, LID technique is combined with site design to form optimal site design [12] and protective design [13]. In New Zealand, the application of LID technique is called low impact urban design and development [14]. With the rapid development and extensive application of computer technique, the related simulation techniques are becoming more mature. The SWMM (storm water management model) is a commonly used drainage system model, which is often used to simulate and evaluate floods of urban areas. The control module of LID has been added to quantify the results of LID simulation in the current version of SWMM (SWMM5.1) [15]. Debusk [16] found that biological-retention could reduce surface runoff by 97% to 99% through the practical research. Qin et al. [17] conducted SWMM simulation of different types of rainfall in Guang-Ming New District, Shenzhen, China, and found that LID caused greater reduction rate of runoff for short-duration rainfall compared to long-duration rainfall. Chui et al. [18] analyzed the cost of different LID types and found that the design scheme was influenced by many factors such as peak flow reduction targets and rainfall types. Rushton [19] used rain gardens and impermeable pavements to reduce the runoff in the study area by about 30%. Cipolla et al. [20] conducted a long-term hydrological simulation experiment of green roofs through SWMM, the results confirmed that green roofs function well in restoring natural water. Martin-Mikle et al. [21]

developed the LID sitting framework on the 666-km$^2$ watershed area, they found that priority sites of LID facilities can lessen the negative effects of urbanization. These studies are mainly focus on the overall system. However, analysis on local ponding regions has seldom been addressed in the literature, yet. [16–21].

In this study, four LID scenarios (① no LID technique. ② LID technique based on infiltration. ③ LID technique based on water storage. ④ LID technique based on the combination of infiltration and water storage) are set up to analyze hydrological characteristics in various rainfall conditions. Concretely, the paper focused on: (1) SWMM model construction and validation by Comprehensive Runoff Coefficient (CRC) method; (2) the design of four LID scenarios; (3) evaluation of four LID scenarios in runoff reduction and peak reduction; (4) the influence of various LID scenarios on local ponding regions; and (5) hydrological characteristics of various LID scenarios under different rainfall amounts. The results of this study can provide some technical support for the construction of drainage systems in urban areas.

## 2. Methods and Materials

### 2.1. SWMM

In this paper, the SWMM is applied to simulate the forming process of urban flooding. SWMM is a dynamic rainfall–runoff simulation model which is primarily used to simulate urban single precipitation event or long-term water quantity and water quality [15].

SWMM generalizes the drainage system into four modules: atmosphere, land surface, groundwater, and transportation. Firstly, the research area is divided into several sub-catchments according to the type of underlying surface. Besides, the runoff process in each sub-catchment is calculated under the specific storm intensity. Each sub-catchment can be divided into three subareas: an impervious area with depression storage ($A_1$), an impervious area without depression storage ($A_2$), and a previous area ($A_3$). The overland flow is calculated by generalizing each subarea into a nonlinear reservoir model. There are three kinds of water routing models in SWMM: steady flow routing model, kinematic wave routing model and dynamic wave routing model. The dynamic wave routing model is selected in this project to simulate the inflow, outflow and reflow in the pipeline [15].

### 2.2. LID

LID is a decentralized small-scale measure module which is included in SWMM. It is environmentally friendly, easy to construct, small in size, economical, and ornamental as landscape [22]. There are five types of LID controls in SWMM: biological mitigation factors such as green vegetation, slit infiltration, porous pavements, rain barrels for rainwater on the roof, and vegetated swamps. Particularly, rain gardens [23], green roofs [24,25], permeable pavements [26,27], vegetative swales [28], and bio-retention cells are used frequently.

In SWMM, several LID modules are created, then added to the corresponding subarea by changing parameters according to the actual situation. Based on the principle of water balance, the SWMM calculates real-time inflow and outflow of the subarea [15]. In this paper, four LID scenarios are adopted in SWMM to compare their runoff under different rainfall patterns.

### 2.3. Study Area

Sucheng District is located in the north of Jiangsu Province, China, between 118°10′07″~118°33′88″ E and 33°47′25″~34°1′16″ N, in the east of Suyu District. It is the political, economic, cultural, and transportation center of Suqian City [29]. The study area is a part of Sucheng District and approximately 260 hectares, as shown in Figure 1a. It is in the transition zone from subtropical to warm temperate zone, with obvious monsoon, transition, and instability characteristics. The annual rainfall and temperature vary greatly. Due to the impact of the monsoon circulation and typhoon in the offshore area, cold and warm air meet frequently, and natural disasters—such as floods—occur frequently.

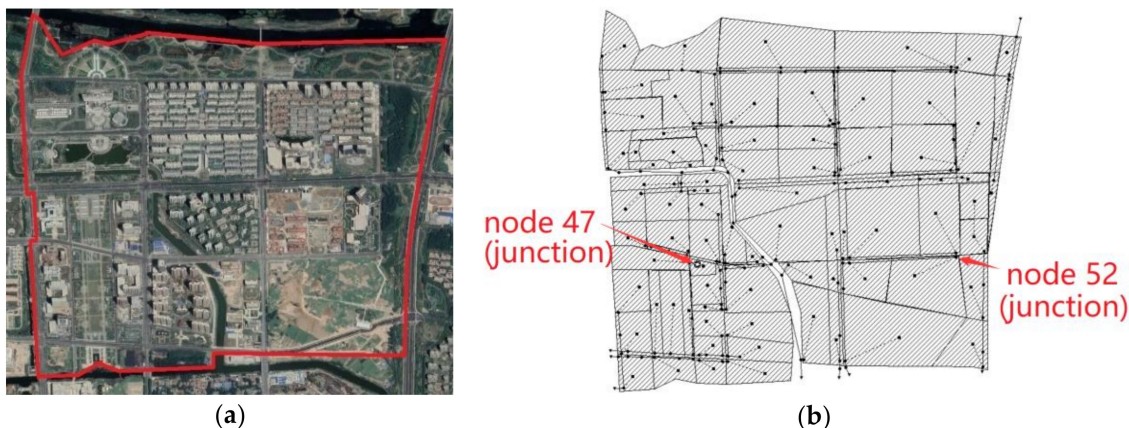

<table>
<tr><td>(**a**)</td><td>(**b**)</td></tr>
</table>

**Figure 1.** Situation in the study area: (**a**) Plan of the study area; (**b**) Drainage system in the base case.

## 3. Modeling Process of SWMM

### 3.1. Sub-Catchment Division

The study area is first divided into several large-scale sub-catchments according to the river course and main avenues, then further subdivided based on side streets and drainage pipe networks. The study area is divided into 83 sub-catchments in total, the river channel in the area is treated as a pipeline, the layout of the drainage system is shown in Figure 1b.

### 3.2. Design Rainstorm

The designing of rainstorm is the basis of the urban rainwater pipe network planning. So far, many well-known scholars have proposed their own storm model [30]. In 1957, Keifer and Chu put forward to the famous Chicago Rainfall Model, an uneven rainfall model based on the intensity–duration–frequency relationship. The Chicago Rainfall Model is widely used in urban water resources engineering. For most urban drainage network designs, Equation (1) is generally adopted [31].

$$i = \frac{q}{167} = \frac{A_1(1 + C\lg P)}{(t_d + b)^c},\tag{1}$$

where $i$ is the average rainfall intensity (mm/min); $q$ is the average rainfall intensity (L/s·ha); $P$ is the return period of design rainfall (a); $t_d$ is the rainfall duration (min); $A_1$, $C$, $b$, $n$ are local empirical parameters. In the designing of the urban drainage network, the time-to-peak ratio $r(0 < r < 1)$ is generally introduced to describe the specific time of the peak flow: the smaller the value of $r$, the closer the peak flow is to the rainfall starting time. In this project, the relationship is summarized as Equation (2) [32]. The duration curve of storm intensity is shown in Figure 2.

$$i = \frac{61.2(1 + 1.05\lg P)}{(t_d + 39.4)^{0.996}},\tag{2}$$

where $A_1 = 61.2$ mm, $b = 39.4$, $C = 1.05$, $n = 0.996$, $r = 0.4$. The return period of flood control standard of the river in this study area is five years, thus the design storm events have return periods of one, three, and five years. The rainfall durations are 120 min and 180 min, which have been recommended in relevant literature [29,32].

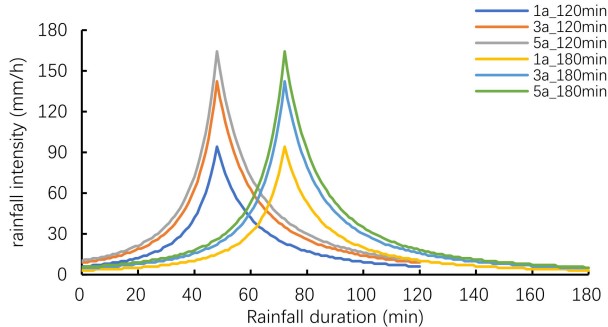

**Figure 2.** Duration curve of storm intensity.

### 3.3. Calibration and Validation of Parameters under Different Rainfall Conditions

The process of calibration is shown in Figure 3. We determined initial parameter values by empirical values recommend in the SWMM manual [15], as shown in the fifth column of Table 1. Then the parameter values were calibrated and validated by the comprehensive runoff coefficient (CRC) method, as shown in Table 2 and Figure 3 [33]. We modified the initial parameter values to meet the corresponding comprehensive runoff coefficient. When we used parameters in the sixth column of Table 1, the corresponding comprehensive runoff coefficients were obtained, as shown in Table 3. The simulation results all satisfied the requirement of the comprehensive runoff coefficient in the densely built central area (0.6~0.8).

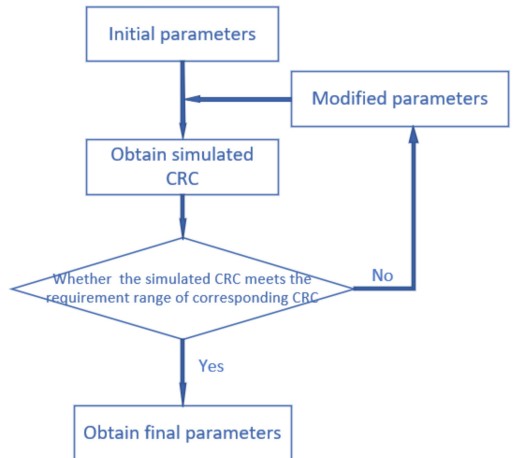

**Figure 3.** Flow chart of the calibration.

**Table 1.** Calibration results of parameters in SWMM.

| No. | Parameter | Parameter Meaning | Range | Initial Value | Final Value |
|-----|-----------|-------------------|-------|---------------|-------------|
| 1 | N-Imperv | Manning coefficients in impervious areas | 0.006~0.05 | 0.012 | 0.012 |
| 2 | N-Perv | Manning coefficients in pervious areas | 0.08~0.5 | 0.12 | 0.1 |
| 3 | Manning-N | Manning coefficient of the pipeline | 0.011~0.24 | 0.012 | 0.013 |
| 4 | S-Imperv | Depression storage in impervious areas/mm | 0.2~5 | 3.1 | 3.2 |
| 5 | S-Perv | Depression storage in pervious areas/mm | 2~10 | 6.1 | 6.6 |
| 6 | Max-Rate | Maximum infiltration rate (mm/h) | 25~75 | 75 | 75 |
| 7 | Min-Rate | Minimum infiltration rate (mm/h) | 0~10 | 4.21 | 3.81 |
| 8 | Decay | Infiltration decay constant (1/h) | 2~7 | 3 | 3 |

**Table 2.** Empirical values of regional comprehensive runoff coefficient.

| Regional Type | Comprehensive Runoff Coefficient |
|---|---|
| Densely built central area | 0.6~0.8 |
| Densely built residential area | 0.5~0.7 |
| Sparsely built residential area | 0.4~0.6 |
| Sparsely populated area | 0.3~0.5 |

**Table 3.** Comprehensive runoff coefficient in various conditions.

|  | T = 120 min | T = 180 min |
|---|---|---|
| P = 1a | 0.610 | 0.627 |
| P = 3a | 0.729 | 0.742 |
| P = 5a | 0.763 | 0.775 |

Note: T is rainfall duration, P is return period, 1a is one year.

### 3.4. Design LID Scenarios

There are four types of LID scenarios: no LID technique, LID technique based on infiltration, LID technique based on water storage, LID technique based on the combination of infiltration and water storage. Figure 4 shows illustration of the distribution of LID facilities.

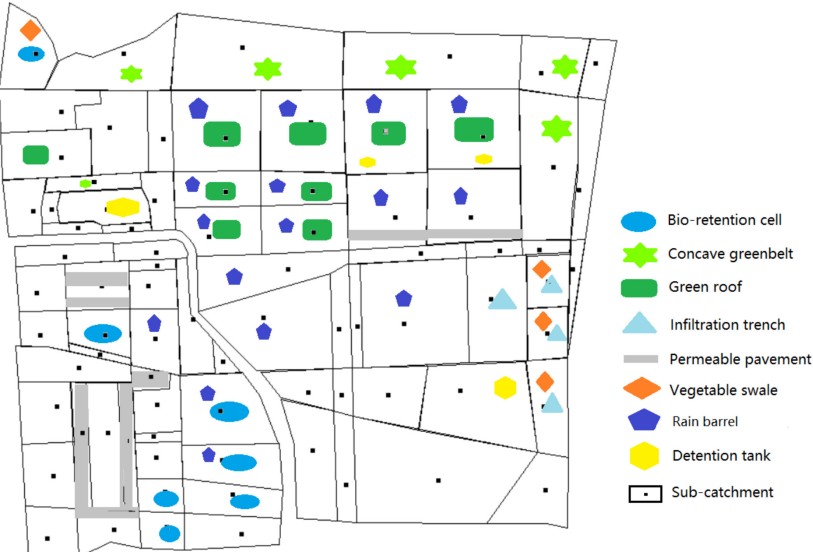

**Figure 4.** Illustration of the distribution of LID facilities.

① No LID technique (noted as Base case), i.e., it does not consider effects of LID. Land use types are shown in Table 4.

**Table 4.** Area of different land use types.

| No. | Land Use Type | Impervious (%) | Surface Area (ha) | Percentage of Occupation (%) |
|---|---|---|---|---|
| 1 | Parking landscape | 40 | 19.6 | 7.54 |
| 2 | Greenland | 30 | 23.7 | 9.12 |
| 3 | Construction area | 80 | 168.2 | 64.69 |
| 4 | Square | 70 | 19.4 | 7.46 |
| 5 | Concrete pavement | 70 | 24.8 | 9.54 |
| 6 | River | - | 4.3 | 1.65 |
| Total | - | - | 260 | 100 |

② LID technique based on infiltration (noted as LID_Infiltrate). The scenario consists of green roofs, concave greenbelts, bio-retention cells (rain gardens), permeable pavements, infiltration trenches, and vegetative swales. Based on land types of the underlying surface, we propose the designing principle: deploying green roofs, rain gardens, and concave greenbelts in densely populated communities; deploying permeable pavements on pedestrian roads, parks and squares; placing infiltration trenches and vegetative swales in the low-lying part of the eastern area. Details are shown in Tables 5 and 6 and Figure 4.

**Table 5.** Area of different LID types.

| No. | LID Type | Surface Area (ha) | Percentage of Occupation (%) |
|-----|----------|-------------------|------------------------------|
| 1 | Green roof | 21.46 | 28.85 |
| 2 | Concave greenbelt | 23.97 | 32.23 |
| 3 | Bio-retention cell | 10.32 | 13.87 |
| 4 | Permeable pavement | 16.69 | 22.44 |
| 5 | Infiltration trenches | 0.36 | 0.49 |
| 6 | Vegetable swale | 1.58 | 2.12 |
| 7 | Total | 74.38 | 100 |

**Table 6.** Partial permeability parameters of LID.

| Layer | Parameter | Unit | Green Roof | Rain Garden | Bio-Retention Cell |
|-------|-----------|------|------------|-------------|--------------------|
| Surface | Berm Height | mm | 50 | 150 | 150 |
| | Vegetation Volume Fraction | - | 0.2 | 0.1 | 0.05 |
| | Surface's roughness | - | 0.13 | 0.12 | 0.12 |
| | Surface Slope | percent | 1 | 0.3 | 0.1 |
| Pavement | Thickness | mm | - | - | - |
| | Void radio | - | - | - | - |
| | Permeability | mm/h | - | - | - |
| Soil | Thickness | mm | 200 | 500 | 500 |
| | Porosity | - | 0.5 | 0.3 | 0.5 |
| | Field capacity | - | 0.3 | 0.2 | 0.2 |
| | Wilting point | - | 0.1 | 0.1 | 0.07 |
| | Conductivity | mm/h | 700 | 500 | 110 |
| Drainage mat | Thickness | mm | 100 | - | - |
| | Void fraction | - | 0.43 | - | - |
| | Roughness | - | 0.03 | - | - |
| Storage | Thickness | mm | - | - | 260 |
| | Void radio | - | - | - | 0.75 |
| | Seepage rate | mm/h | - | 200 | 80 |
| Drain | Flow coefficient | - | - | - | 0 |
| | Flow exponent | - | - | - | 0.5 |
| | offset height | mm | - | - | 150 |

**Table 6.** *Cont.*

| Layer | Parameter | Unit | Permeable Pavement | Infiltration Trench | Vegetative Swale |
|---|---|---|---|---|---|
| Surface | Berm Height | mm | 25 | 150 | 200 |
| | Vegetation Volume Fraction | - | 0 | 0 | 0.1 |
| | Surface's roughness | - | 0.12 | 0.24 | 0.13 |
| | Surface Slope | percent | 1 | 1 | 0.8 |
| Pavement | Thickness | mm | 60 | - | - |
| | Void radio | - | 0.13 | - | - |
| | Permeability | mm/h | 200 | - | – |
| Soil | Thickness | mm | 150 | - | - |
| | Porosity | - | 0.5 | - | - |
| | Field capacity | - | 0.1 | - | - |
| | Wilting point | - | 0.024 | - | - |
| | Conductivity | mm/h | 100 | - | - |
| Drainage mat | Thickness | mm | - | - | - |
| | Void fraction | - | - | - | - |
| | Roughness | - | - | - | - |
| Storage | Thickness | mm | 250 | 600 | - |
| | Void radio | | 0.43 | 0.75 | - |
| | Seepage rate | mm/h | 600 | 24 | - |
| Drain | Flow coefficient | - | 0.69 | 0.69 | - |
| | Flow exponent | - | 0.5 | 0.5 | - |
| | offset height | mm | 6 | 6 | - |

③ LID technique based on water storage (noted as LID_Storage). Rain barrels are located in densely populated areas. Each rain barrel accounts for about 0.2% of each selected subarea. Storage units are set up for runoff control in sub-catchment areas which are prone to flooding. As shown in Tables 7 and 8 and Figure 4.

**Table 7.** Partial storage parameters of LID.

| Layer | Parameter | Unit | Rain Barrel | Storage Unit | | | |
|---|---|---|---|---|---|---|---|
| | | | | Sto1 | Sto2 | Sto3 | Sto4 |
| Storage | Height | mm | 800 | 4300 | 2000 | 2000 | 3000 |
| | Bottom area | sqm | 0.58 | 14,400 | 2000 | 150 | 3000 |
| Drain | Flow coefficient | - | 0.68 | - | - | - | - |
| | Flow exponent | - | 0.5 | - | - | - | - |
| | offset height | mm | 125 | - | - | - | - |
| | Drain delay | hour | 5.8 | - | - | - | - |

④ LID technique based on the combination of infiltration and water storage (noted as LID_Combination). This scenario is a combination of scenario ② and scenario ③, parameters are shown in Tables 5–8.

**Table 8.** Area of different LID types.

| No. | LID Type | Volume (m³) | Number | Total Volume (m³) |
|-----|----------|-------------|--------|-------------------|
| 1 | Rain barrel | 0.464 | 16 | 7.424 |
| 2 | Storage unit 1 | 61,920 | 1 | 61,920 |
| 3 | Storage unit 2 | 4000 | 1 | 4000 |
| 4 | Storage unit 3 | 300 | 1 | 300 |
| 5 | Storage unit 4 | 9000 | 1 | 9000 |
| Total | - | - | 20 | 75,227.424 |

## 4. Results

### 4.1. Rainfall–Runoff Relationship of Various LID Scenarios

Figure 5 shows the relationship between rainfall and runoff under different LID scenarios. In the rainfall duration of 120 min, the base time of runoff concentration is about 240 min; in the rainfall duration of 180 min, the base time of runoff concentration is about 360 min. It can be seen that the peak time of rainfall is roughly the same as the peak time of flow with slightly delay. The reason is that the underlying surface always has water storage units and other facilities with the function of storing and infiltrating water. Tables 9 and 10 show the reduction rate of peak flow and flood volume with various types of LID, respectively.

**Table 9.** Reduction rates of peak flow by various LID types.

| Rainfall Duration | Return Period | LID_Storage | LID_Infiltrate | LID_Combination |
|-------------------|---------------|-------------|----------------|-----------------|
| 120 min | 1a | 1.44% | 30.55% | 32.44% |
|  | 3a | 1.45% | 28.61% | 30.63% |
|  | 5a | 1.43% | 28.25% | 30.25% |
| 180 min | 1a | 1.46% | 30.46% | 32.51% |
|  | 3a | 1.45% | 28.36% | 30.38% |
|  | 5a | 1.39% | 28.09% | 30.01% |

**Table 10.** Reduction rates of flood volume by various LID types.

| Rainfall Duration | Return Period | LID_Storage | LID_Infiltrate | LID_Combination |
|-------------------|---------------|-------------|----------------|-----------------|
| 120 min | 1a | 1.46% | 29.76% | 31.80% |
|  | 3a | 1.33% | 29.41% | 31.24% |
|  | 5a | 1.28% | 29.35% | 31.15% |
| 180 min | 1a | 1.45% | 29.75% | 31.77% |
|  | 3a | 1.29% | 29.46% | 31.27% |
|  | 5a | 1.26% | 29.38% | 31.16% |

In the above rainfall–runoff diagrams, the surface runoff hydrograph of the Base case and LID_Storage almost coincide. When the rainfall duration is 120 min, the return periods are 1a, 3a, 5a, the reduction rates of peak flow in LID_Combination scenario are 32.44, 30.63, and 30.25%, respectively; and when the rainfall duration is 180 min, the reduction rates are 32.51, 30.38, and 30.01%. It shows that with a certain rainfall duration, the reduction rate of the peak flow is weaker when the rainfall intensity is greater. For the case with different rainfall durations but the same rainfall return period, the reduction rate of peak flow caused by infiltration will be slightly weakened in 180 min compared with 120 min. When the rainfall duration is 120 min, the return periods are 1a, 3a, 5a, the reduction rates of flood volume in LID_Combination scenario are 31.80, 31.24, and 31.15%, respectively; and when the rainfall duration is 180 min, the reduction rates are 3.77, 31.27, and 31.16%. It shows that the reduction rate of the flood volume is weaker when the rainfall intensity is greater. Among the 18 results, the maximum reduction rate of peak flow can reach up to 32.51% (T = 180 min,

1a, LID_Combination), the minimum is only 1.39% (T = 180 min, 5a, LID_Storage). The maximum reduction rate of total flood volume can reach up to 31.8% (T = 120 min, 1a, LID_Combination), the minimum is only 1.26%. In general, the efficiency of runoff reduction is ranked as follows: LID_Combination > LID_Infiltrate > LID_Storage > Base case. Xie et al. [27] have similar results in Nanshan village's research, with the maximum reduction rate of peak flow 100% (T = 180 min, 0.33a, LID_Combination), the minimum of 8.19% (T = 180 min, 5a, LID_Grassed Swale); and the maximum reduction rate of total flood volume 100% (T = 180 min, 0.33a, LID_Combination), the minimum of 6.47% (T = 180 min, 5a, LID_Grassed Swale).

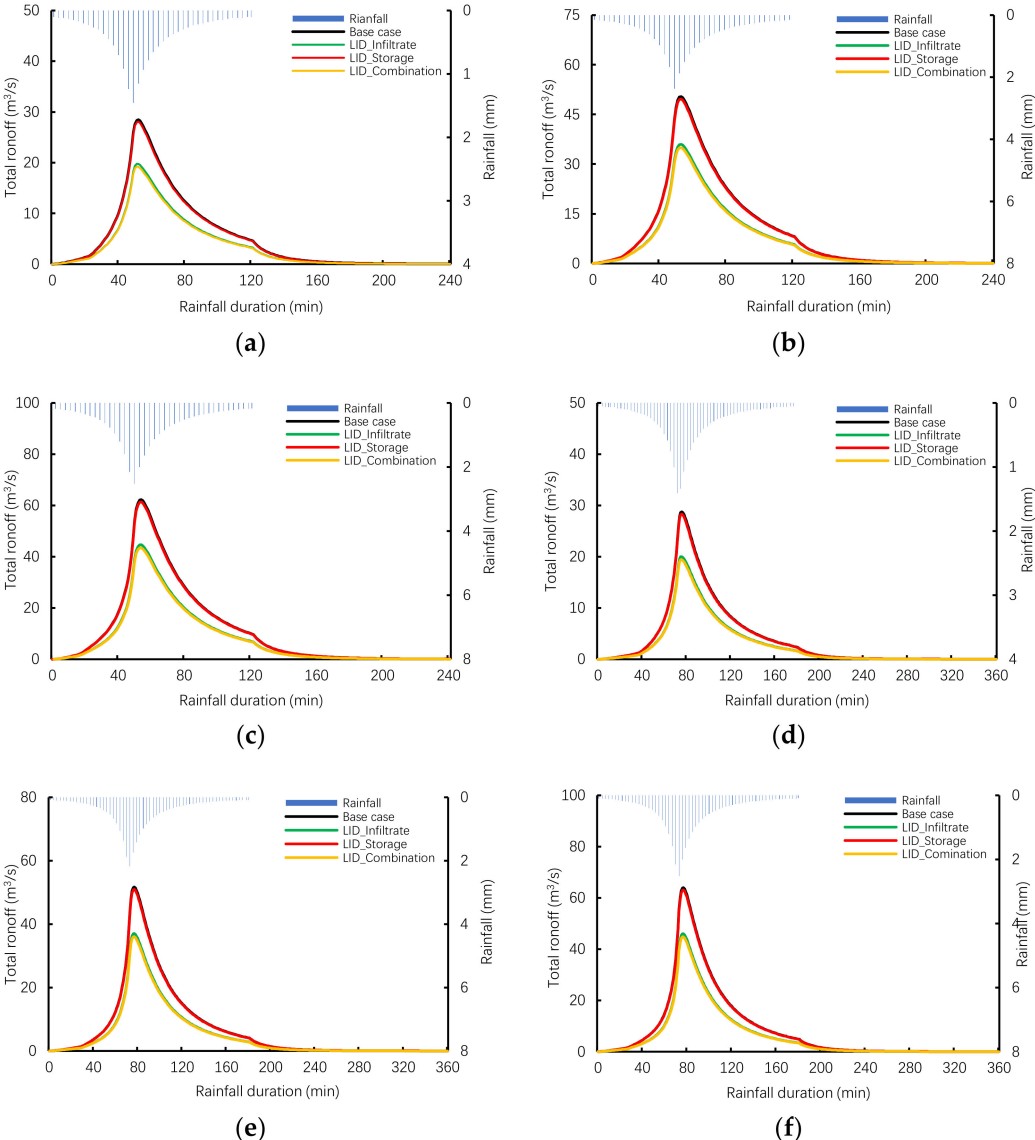

**Figure 5.** Rainfall–runoff relationship under different LID scenarios: (**a**) Rainfall duration is 120 min, return period is 1a; (**b**) Rainfall duration is 120 min, return period is 3a; (**c**) Rainfall duration is 120 min, return period is 5a; (**d**) Rainfall duration is 180 min, return period is 1a; (**e**) Rainfall duration is 180 min, return period is 3a; (**f**) Rainfall duration is 180 min, return period is 5a.

In a word, the peak flow decreases gradually with the shorter rainfall duration and weaker rainfall intensity. The flood hydrograph is characteristic of the rapid rise and mitigatory recession with decreased flood volume and prolonged flood duration. Results in the study suggest that infiltration facilities reduce the surface runoff greatly while storage facilities have little effect on it. Retention tanks

are more likely to mitigate flooding in local regions and the size of water storage facilities are not large enough. However, infiltration facilities usually consist of a series of porous structures connected to the outside air, allow rainwater to be infiltrated directly and reduce surface runoff at the source.

### 4.2. Influences of Various Types of LID on Local Ponding Regions

There have been few studies on local nodes, this paper analyzes and discusses nodes with flood, so as to improve the reliability of the conclusion. Since node 47 (junction node) and node 52 (junction node) are two nodes with the longer ponding time and more ponding depth in Base case, they are selected as typical ponding nodes for analysis and discussion, as shown in Figure 1b. Influences of different types of LID on local ponding regions are shown in Figures 6 and 7, respectively.

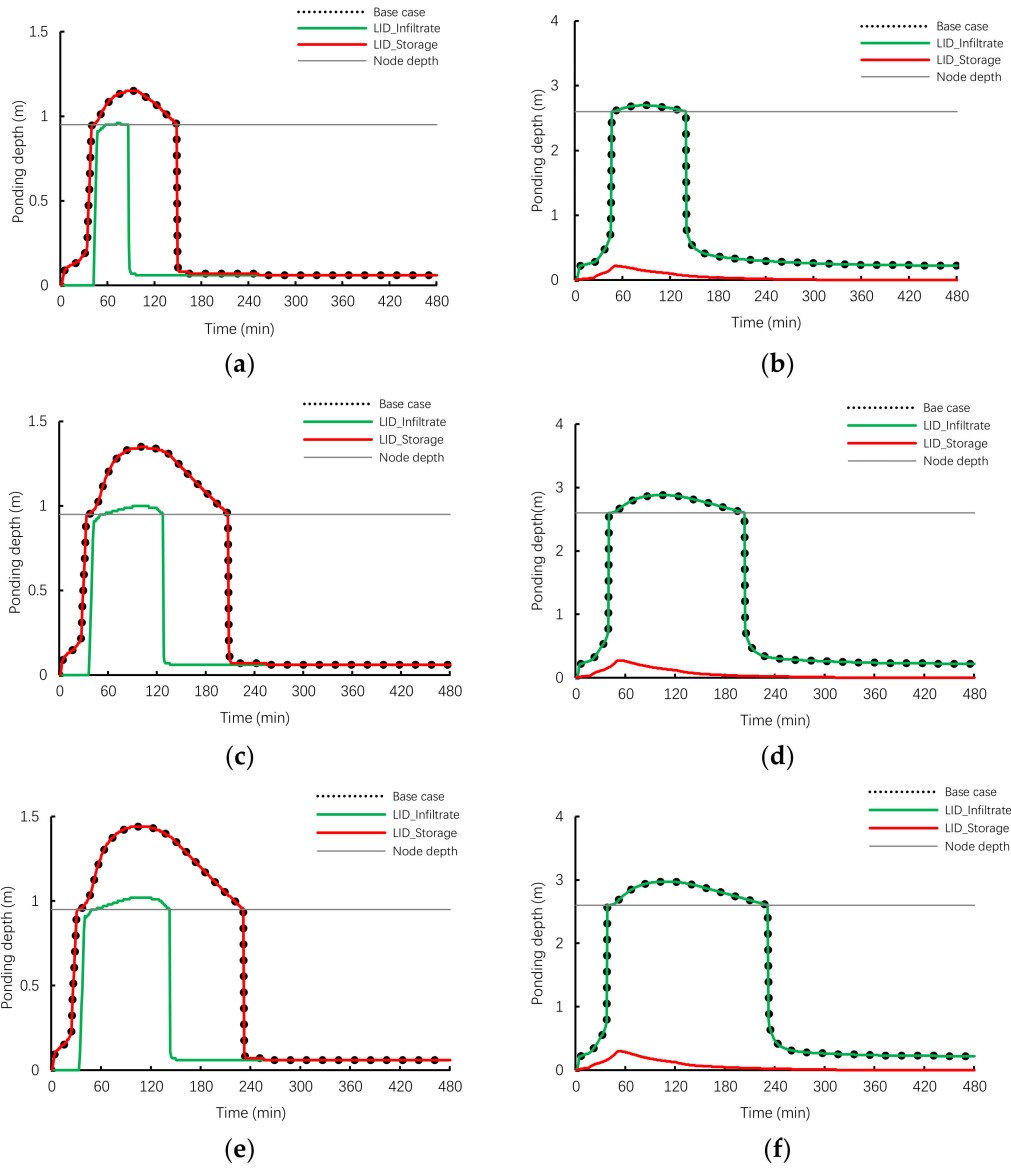

**Figure 6.** Ponding time of various LID scenarios when rainfall duration is 120 min: (**a**) When return period is 1a, ponding time at node 47; (**b**) When return period is 1a, ponding time at node 52; (**c**) When return period is 3a, ponding time at node 47; (**d**) When return period is 3a, ponding time at node 52; (**e**) When return period is 5a, ponding time at node 47; (**f**) When return period is 5a, ponding time at node 52.

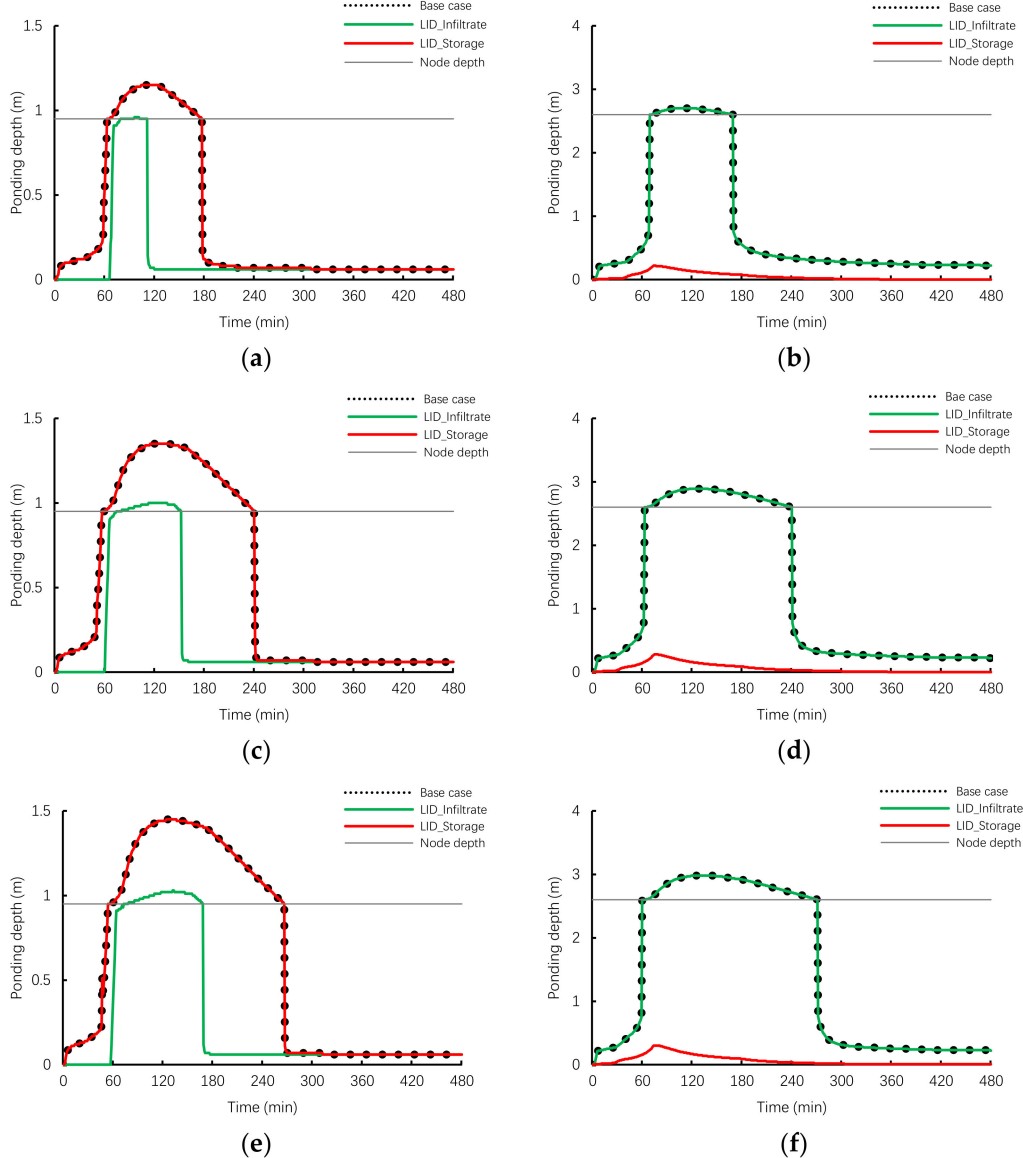

**Figure 7.** Ponding time of various LID scenarios when rainfall duration is 180 min: (**a**) When return period is 1a, ponding time at node 47; (**b**) When return period is 1a, ponding time at node 52; (**c**) When return period is 3a, ponding time at node 47; (**d**) When return period is 3a, ponding time at node 52; (**e**) When return period is 5a, ponding time at node 47; (**f**) When return period is 5a, ponding time at node 52.

We determined the ponding area based on the DEM (digital elevation model) and questionnaires given to local residents. The ponding area at node 47 is 734 m$^2$, and at node 52 is 7850 m$^2$. It can be seen that for ponding water at node 47, ponding depth curves of Base case coincide with LID_Storage, and ponding depth curves of LID_Infiltrate are obviously lower than that of the base case. Ponding time varies greatly with different rainfall patterns. When the rainfall durations are 120 min and 180 min, the minimum ponding time at node 47 are 29.4 min, 31.8 min (1a, LID_Infiltration); the maximum ponding time are 201 min and 212.4 min (5a, base case), respectively. When the rainfall duration is 120 min and return periods are 1a, 3a, and 5a, reduction rates of water ponding time at node 47 are 73.1, 55.7, and 53.1%, respectively. When rainfall lasts for 180 min, reduction rates are 72.3, 57.3, and 54.5% respectively. As for ponding water at node 52, ponding depth curves of base case coincide with LID_Infiltrate. The ponding depth curve of LID_Storage is obviously lower than that of the base case. When there are storage facilities, there is no ponding water at node 52.

Infiltration facilities are more effective in flood reduction at node 47, storage facilities mitigate floods dramatically at node 52, and storage facilities perform much better in local regions than infiltration facilities. The flood hydrograph in LID_Infiltrate scenario is characterized by the rapid rise and recession at node 47, but there is a more mitigatory flood hydrograph in LID_Storage scenario at node 52. As the rainfall intensity decreases and rainfall duration increases, peak time is delayed. The ponding time becomes longer with increasing rainfall intensity, and the reduction rate of ponding time is smaller with increasing rainfall intensity.

### 4.3. Characteristics of Hydrology under Different Rainfall Amounts

The hydrological characteristics of different types of LID facilities under different rainfall amounts are shown in Figure 8. Curves of the base case basically coincide with LID_Storage, LID_Infiltrate basically coincide with LID_Combinarion. Infiltration facilities have a greater reduction effect on runoff as the total amount of rainfall increases.

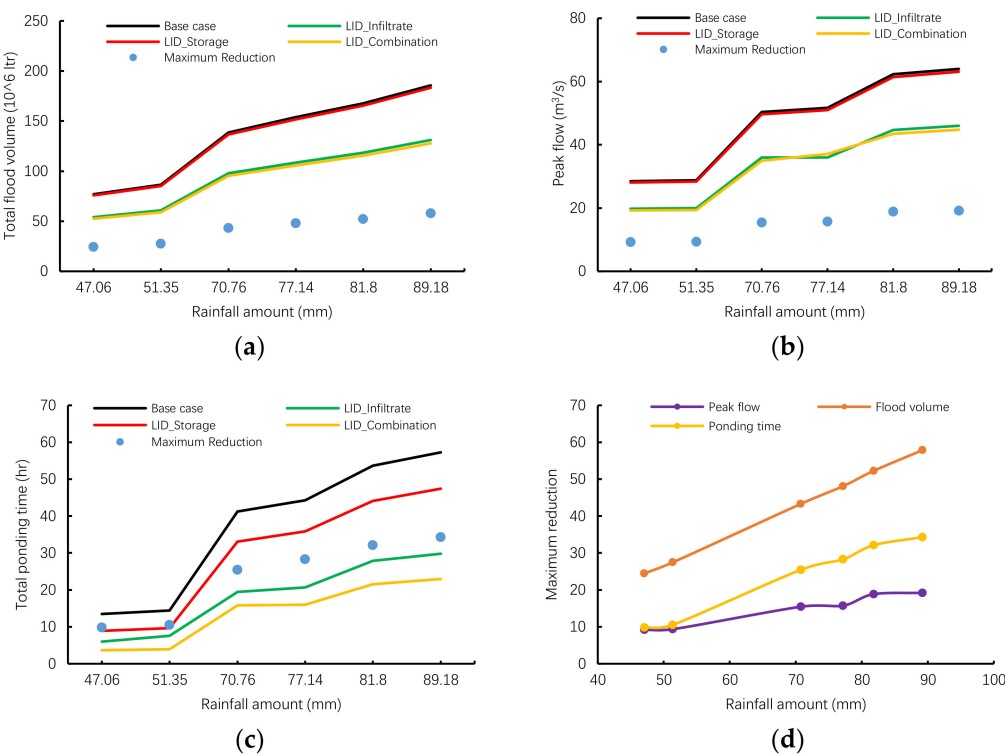

**Figure 8.** Relationship between hydrological characteristics and rainfall amounts: (**a**) Relationship between flood volume and rainfall amounts; (**b**) Relationship between peak flow and rainfall amounts; (**c**) Relationship between total ponding time and rainfall amounts; (**d**) Relationship between maximum reduction and rain amounts of various hydrological characteristics.

Runoff reduction varies greatly with rainfall amounts. When the rainfall amount is 47.06 mm, the maximum reduction of flood volume during the continuous rainfall–runoff process (240 min) is 24,470 m$^3$. When the rainfall amount is 89.18 mm, the maximum reduction of flood volume can reach 57,850 m$^3$ during the continuous rainfall–runoff process (360 min). When the rainfall amount is 47.06 mm, the maximum reduction of peak flow is 9.24 m$^3$/s. When the rainfall amount is 89.18 mm, the maximum reduction of peak flow is 19.2 m$^3$/s. When the rainfall amount is 47.06 mm, the maximum reduction time is 13.48 h. When the rainfall amount is 89.18 mm, the maximum reduction time can reach 57.26 h. As shown in Figure 8a–c. Under all the LID designs, runoff reduction gradually increases with the increasing rainfall amount, peak reduction becomes stable when rainfall amount reaches

about 81.8 mm. As shown in Figure 8d. In general, maximum reduction rates are ranked as follows: flood volume > ponding time > peak flow.

## 5. Discussion and Conclusions

This paper focused on the effect of LID techniques on flood control in typical urban areas of China. Specifically, we simulated the effect of four LID types on peak flow and flood volume under different rainfall conditions. The reduction effect of different LID types on ponding time in ponding regions was also discussed. We conclude this work as below:

(1) For the whole system, as the flood volume is reduced and flood duration is prolonged, the peak flow decreases dramatically and the flood hydrograph is characteristic of the rapid rise and mitigatory recession. Infiltration facilities can significantly reduce surface runoff, LID_Combination works best. All the LID designs are more effective in runoff reduction during shorter and heavier rainfall events. Furthermore, compared to the rapid rise and recession of flood hydrograph with conventional flood control measures, the flood hydrograph is more mitigatory after adding LID designs.

(2) For local flooding regions, infiltration facilities perform better at node 47 while storage facilities perform better at note 52. Storage facilities perform much better in local regions than infiltration facilities. Furthermore, there is a more mitigatory flood hydrograph in LID_Storage scenario at node 52 compared with the LID_Infiltrate scenario at node 47.

(3) Runoff reduction increases with the increasing rainfall amount, but the reduction rate gradually decreases with the increase of rainfall amount. Peak reduction becomes stable when rainfall amount reaches about 81.8 mm. Maximum reduction rates are ranked as follows: flood volume > ponding time > peak flow. If we take more rainfall conditions into consideration and conduct more experiments, there should be more clear and credible results.

(4) The simulated results can provide the arrangement of LID facilities in Sucheng District of Suqian City. For example, when there is a low-lying impermeable area, storage unit should be the first priority; when there is a large-scale square with concrete or asphalt pavements, permeable pavement may be the better choice. Additionally, we can combine the cost of each type of LID techniques, comprehensively considering various factors such as real-time rainfall in the field, getting a more suitable scheme to provide integrated technical support references for the sponge city construction.

**Author Contributions:** Y.B. designed and conducted the experiments. R.Z. collected the data. N.Z. organized the paper and analyzed the data. Y.B. wrote the paper. X.Z. provided useful advice and checked the paper for revisions and grammar.

**Funding:** This research has been financially supported by the National Key Research and Development Program (2016YFC0401005, 2016YFC0401004), and the National Natural Science Foundation of China (91547208), and the Funds for the Central Universities, HUST (2016YXZD046, 2017KFYXJJ191).

**Conflicts of Interest:** The authors declare no conflict of interest.

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
