# Peer review of "Storm Water Management of Low Impact Development in Urban Areas Based on SWMM"

_water, doi:10.3390/w11010033_

Round 1

Reviewer 1 Report

The paper discusses an important issue about the effectiveness  of  LID systems in different conditions. However, in its current form the scientific contribution of the paper is not well defined and the novelty is very limited. Although the case study has potential, the results describe well known facts. I would recommend to clearly articulate the research aim and refine the scientific contribution. 

Major comments:

The literature review identifies 3 research gaps. 

1)  only a small area is considered

2)  LID systems have not been considered appropriately

3)  The studies takes the overall performance into account and not local details.

However, these research gaps are currently not underpinned by the literature review section. The authors should also expand the scope of their review e.g. tools like SUSTAIN developed by the USEPA and other tools that have been developed and applied to address a range of issues identified by the authors [1]. Further the research aim of the paper is not clearly defined. 

[1] Chelsea J. Martin-Mikle, Kirsten M. de Beurs, Jason P. Julian, Paul M. Mayer, Identifying priority sites for low impact development (LID) in a mixed-use watershed, Landscape and Urban Planning, Volume 140, 2015

Author Response

Answer 1These research gaps are summarized based on small-scale literatures, we are very sorry for not citing this reference before. Thank you for the recommended reference [1]. It’s very necessary to add it in the introduction. We improved these research gaps in the introduction.

“Martin-Mikle et al [21] developed the LID sitting framework on the 666-km2 watershed area, they found that priority sites of LID facilities can lessen the negative effects of urbanization.”

 “These studies are mainly focus on the overall system. However, analysis on local ponding regions has seldom been addressed in the literature, yet. [16-21].”

Answer 2We modified the introduction to make the aim of the paper clearly understand and refine the scientific contribution. We improved the discussion and conclusion in details.

“In this study, four LID scenarios (no LID technique. LID technique based on infiltration. LID technique based on water storage. LID technique based on the combination of infiltration and water storage.) are set up to analyze hydrological characteristics in various rainfall conditions. Concretely, the paper focused on: (1) SWMM model construction and validation by Comprehensive Runoff Coefficient (CRC) method; (2) the design of four LID scenarios; (3) evaluation of four LID scenarios in runoff reduction and peak reduction; (4) the influence of various LID scenarios on local ponding regions; and (5) hydrological characteristics of various LID scenarios under different rainfall amounts. The results of this study can provide some technical support for the construction of drainage systems in urban areas.”

Thank you very much for your useful comments and suggestions!

 [1] Chelsea J. Martin-Mikle, Kirsten M. de Beurs, Jason P. Julian, Paul M. Mayer, Identifying priority sites for low impact development (LID) in a mixed-use watershed, Landscape and Urban Planning, Volume 140, 2015

Reviewer 2 Report

The aim of the study must be clearly precised.      

Fig 2b, please use larger font

Equation 1&2 – there is sth wrong with      numbers implemented. Please check

What was a reason of selection of the      rainfalls?

What was the reason of selection durations?

 How      parameters in sections 3.2.3, 3.2.4 were calibrated?

(1 - base) scenario should be described.      What are the use of area? % of different use

Table 3 and 4 are unclear, please divide      the subsections. Add missed units!

 Figure 4 –      rainfall duration 120 min and 180 minutes should be set in columns

Figure 5 and 6 – I cannot see “base case”

The selection of scenarios in unclear.      Reader is not able to understand what is included in different scenarios.      E.g. Authors inform that “Based on land types of the underlying surface, we      propose the  designing principle: deploying green      roofs, rain gardens, and low elevation greenbelts in densely  populated      communities; deploying permeable pavements on pedestrian roads, parks and      squares; placing infiltration trenches and vegetative swales in the      low-lying part of the eastern area, partial  permeability parameters      are shown in Table 3.” Ok, but what is the share of those areas, where are they located?

Some of conclusions are obvious.

Author Response

1, The aim of the study must be clearly precise.

Answer: We modified the introduction to make the aim of the paper clearly understand and refine the scientific contribution.

“In this study, four LID scenarios (no LID technique. LID technique based on infiltration. LID technique based on water storage. LID technique based on the combination of infiltration and water storage.) are set up to analyze hydrological characteristics in various rainfall conditions. Concretely, the paper focused on: (1) SWMM model construction and validation by Comprehensive Runoff Coefficient (CRC) method; (2) the design of four LID scenarios; (3) evaluation of four LID scenarios in runoff reduction and peak reduction; (4) the influence of various LID scenarios on local ponding regions; and (5) hydrological characteristics of various LID scenarios under different rainfall amounts. The results of this study can provide some technical support for the construction of drainage systems in urban areas.”

2Fig 2b, please use larger font; Equation 1&2 – there is sth wrong with numbers implemented. Please check

Answer: We have checked and amended the Fig 2b and Equation 1&2.

3, What was a reason of selection of the rainfalls? What was the reason of selection durations?

Answer: The return period of flood control standard of the river in this study area is 5-year, thus the design storm events have return periods of 1-, 3-, and 5-year. We use empirical values of the short-duration (T=120 min, T=180 min).

4, How parameters in sections 3.2.3, 3.2.4 were calibrated?

Answer: We improve this section: 3.3. Calibration and Validation of Parameters under Different Rainfall Conditions. And we add the Flow chart of the calibration.

 “The process of calibration is shown in Figure 4. We determined initial parameter values by empirical values recommend in the SWMM manual [15], as shown in the fifth column of Table 1. Then the parameter values were calibrated and validated by Comprehensive Runoff Coefficient (CRC) method, as shown in Table 2 and Figure 4 [34]. We modified the initial parameter values to meet the corresponding comprehensive runoff coefficient. When we used parameters in the sixth column of Table 1, the corresponding comprehensive runoff coefficients were obtained, as shown in Table 3. The simulation results all satisfied the requirement of the comprehensive runoff coefficient in the densely built central area (0.6~0.8).”

5.1. (1 - base) scenario should be described. What are the use of area? % of different use

5.2 Table 3 and 4 are unclear, please divide the subsections. Add missed units!

5.3 Figure 4 – rainfall duration 120 min and 180 minutes should be set in columns

5.4 Figure 5 and 6 – I cannot see “base case”

5.5 what is the share of those LID areas, where are they located?

Answer: these problems all have been solved in the revised manuscript.

Answer: 5.1-5.5

5.1 Table 4.

5.2 Table 6 and Table 7

5.3 Figure 6

5.4 Figure 7 and Figure 8.

5.5 Figure 5 and Table 5.

6, Some of conclusions are obvious.

Answer: We improved the discussion and conclusion in details.

Thank you very much for providing useful comments and suggestions! It is very necessary to improve these deficiencies.

Reviewer 3 Report

In the paper Authors described   the  effect  of  LID  on  flood  control  in  urban  areas  of  China (Suqian District of Suqian City, Jiangsu Province). Mainly the effect of four LID types on peak flow  and flood volume under different  rainfall durations and rainfall intensities were simulated. The paper is interesting mainly for designers and authorities becouse showes the effect of differents LID configurations on decreasing flood hazard in urban areas. Maior comments:
1. Page 5, line 180-182: Did duration of rainfall equal 120 and 180 min was taken from time of concentration of flow from watershed? Please explain abbreviations: 1a, 3 a .... Does 1a is one event per year?
2. Page 5, Please give more informations about calibration procedure. How were calibrated parameters in table 2? Please give details observed events that were used to calibration. What were measurements of quality models used  during calibration?
3. Page 6 and 7, tanle 3: Please give informations how were assesmend the parameters of LID
4. Page 8, Lack details about used method for calculation of overland flow, channel flow. What were methods used for determine this characteristics? How was hydraulics width and slope assesmend for watersheds?
5. Page 11, chapter 4.2: Please explain how was assesment ponding area? What source informations were used for assesment this parameter?
6. Page 15, line 311-313: May be duration of rainfall is determining this relations? Larger rainfall can have shorter time (higher intensity) but volume of runoff from this event could be lesser. Please to discus this problem.
7. General comments: Please give more emphasis on clearly present the novetly of this study. I think that the conclusions are commonly knows.

Author Response

1. Page 5, line 180-182: Did duration of rainfall equal 120 and 180 min was taken from time of concentration of flow from watershed? Please explain abbreviations: 1a, 3 a .... Does 1a is one event per year?

Answer: The return period of flood control standard of the river in this study area is 5-year, thus the design storm events have return periods of 1-, 3-, and 5-year. We use empirical values of the short-duration (T=120 min, T=180 min). Table 3, note: T is rainfall duration, P is return period, 1a is one event per year.

2. Page 5, Please give more information about calibration procedure. How were calibrated parameters in table 2? Please give details observed events that were used to calibration. What were measurements of quality models used during calibration?

Answer: We improve this section: 3.3. Calibration and Validation of Parameters under Different Rainfall Conditions. And we add the Flow chart of the calibration.

“The process of calibration is shown in Figure 4. We determined initial parameter values by empirical values recommend in the SWMM manual [15], as shown in the fifth column of Table 1. Then the parameter values were calibrated and validated by Comprehensive Runoff Coefficient (CRC) method, as shown in Table 2 and Figure 4 [34]. We modified the initial parameter values to meet the corresponding comprehensive runoff coefficient. When we used parameters in the sixth column of Table 1, the corresponding comprehensive runoff coefficients were obtained, as shown in Table 3. The simulation results all satisfied the requirement of the comprehensive runoff coefficient in the densely built central area (0.6~0.8).”

3. Page 6 and 7, table 3: Please give information how were assessed of the LID parameters

Answer: We determine the parameters of LID based on empirical values.

4. Page 8, Lack details about used method for calculation of overland flow, channel flow. What were methods used for determine these characteristics? How was hydraulics width and slope assessed for watersheds?

Answer: Area = hydraulics width * hydraulics width

5. Page 11, chapter 4.2: Please explain how was assesment ponding area? What source information were used for assesment this parameter?

Answer: We estimate empirical values of the ponding area based on literatures and the terrain. First, we determine the ponding area based on literatures and the terrain, then we use SWMM to obtain simulated ponding depth of the node (junction).

6. Page 15, line 311-313: May be duration of rainfall is determining this relationship? Larger rainfall can have shorter time (higher intensity) but volume of runoff from this event could be lesser. Please to discuss this problem.

Answer: we have improved it in the result and discussion.

7. General comments: Please give more emphasis on clearly present the novelty of this study. I think that the conclusions are commonly knows.

Answer: We improved the discussion and conclusion in details.

Thank you very much for providing useful comments and suggestions!

Round 2

Reviewer 2 Report

Authors improved the manuscript, and the quality of presentation of the methodology and results increased significantly. In my opinion, the manuscript can be accepted for publication

Author Response

Thanks for your time. Sincere best wishes to you !

December 19, 2018.

Reviewer 3 Report

Authors improved paper according to more comments. But I feel slight lack answers on following comments:
Comment 1: Please explain more preciselly what do Authors mean as empirical values of duration time of rainfall? I wonder if this time was assesment on base time of concentration?
Comment 4: In my comment I mean about model used for calculation overland flow (Did Authors use nonlinear reservoir or unit hydrograph?) and chanell flow (stationary on nonstationary flow?)
Comment 5:. What datas about terrain was used to assesment ponding area? DTM?

Author Response

Dear professor,

I am writing to answer the questions that you have proposed in this paper. Thank you very much for your useful comments and suggestions. All the comments and suggestions are valuable. We think it’s very necessary to improve these deficiencies.

Comment 1: Please explain more precislly what do Authors mean as empirical values of duration time of rainfall? I wonder if this time was assesment on base time of concentration?

Answer: The rainfall duration is determined based on relevant literatures, the base time of concentration is determined based on the simulated runoff. In the rainfall duration of 120 min, Figure 5 (a-c) show that there is little or no runoff after 240 min. In the rainfall duration of 180 min, Figure 5 (d-f) show that there is little or no runoff after 360 min. We improved them in the paper in section 3.2 & section 4.1.

In Section 3.2. “The rainfall durations are 120 min and 180 min, which have been recommended in relevant literatures [32-33].

In section 4.1. “In the rainfall duration of 120 min, the base time of runoff concentration is about 240 min; in the rainfall duration of 180 min, the base time of runoff concentration is about 360 min.

Comment 4: In my comment I mean about model used for calculation overland flow (Did Authors use nonlinear reservoir or unit hydrograph?) and channel flow (stationary on nonstationary flow?

Answer: We were very sorry they were not described before. The nonlinear reservoir model is used for the runoff calculation. We choose the nonstationary flow in SWMM. We improved them in section 2.1.

In section 2.1. “The overland flow is calculated by generalizing each subarea into a nonlinear reservoir model. There are three kinds of water routing models in SWMM: steady flow routing model, kinematic wave routing model and dynamic wave routing model. The dynamic wave routing model is selected in this project to simulate the inflow, outflow and reflow in the pipeline [15].

Comment 5: What data about terrain was used to assesment ponding area? DTM?

Answer: First, we used DEM (Digital Elevation Model) to analysis the elevation of each sub-catchment. Then, we made some questionnaires to people who live in Suqian city for a long time. We improved them in section 4.2.

In section 4.2. “We determined the ponding area based on the DEM (Digital Elevation Model) and questionnaires given to local residents. The ponding area at node 47 is 734 m2, and at node 52 is 7850 m2.

Thank you very much for providing useful comments and suggestions! Best regards to you.

December 19, 2018.
